# Immune Cell Interplay in the Fight Against GBM

**DOI:** 10.3390/cancers17050817

**Published:** 2025-02-26

**Authors:** Nico Vallieri, Angeliki Datsi

**Affiliations:** Institute for Transplantation Diagnostics and Cell Therapeutics, Medical Faculty and University Hospital Duesseldorf, Heinrich-Heine-University Düsseldorf, 40225 Duesseldorf, Germany; nico.vallieri@med.uni-duesseldorf.de

**Keywords:** glioblastoma, immunotherapy, tumor-microenvironment, checkpoint inhibitors, clinical trials

## Abstract

Despite multimodal interventions, managing glioblastoma remains challenging. Beyond proliferation, invasion, and evasion, cells are resistant to chemo- and radiotherapy. In GBM, several mechanisms are in play that contribute to the creation and maintenance of a highly immunosuppressive microenvironment, promoting tumor growth and immune escape. Furthermore, the high accumulation of immunosuppressive cells, the pronounced expression of immune checkpoint molecules, and the low number of neoantigens are hallmarks of GBM and contribute to the challenge of therapeutic approaches. In this review, we will discuss how GBM fosters tumor growth and explore treatment options such as small molecules, immunotherapies, and cell therapies, and their potential in clinical trials.

## 1. Introduction

### 1.1. Characteristics of Hot and Cold Tumors

The formation of cancer is based on the development of malignant cell precursors that manage to escape from the immune system and therefore grow uncontrolled into malignant tumors. One of the hallmarks of these abnormal cells is their ability to migrate and therefore invade healthy tissues and even metastasize into neighboring organs [1]. These neoplasms can obtain advantageous phenotypes, which maintain proliferative signals. Additionally they maintain a replicative immortality and the capacity to evade or block the activity of growth suppressors. Their own proliferation causes and shapes a hypoxic environment, which further supports its own survival and growth. Its constant supply of nutrients is achieved by a strong angiogenesis, further supporting the tumors invasion and metastasis [2,3].

In addition to genomic instability leading to neoplastic cells, the second hallmark of cancer is tumor-promoting immune infiltration, followed by evasion of immune system defenses and resistance of newly formed cancer cells to cell death. Although immunosurveillance is an important aspect of cancer, equally critical is the characterization of the tumor type itself [1]. Immunosurveillance is a critical mechanism by which the immune system detects and eliminates neoplastic cells and pathogens to maintain cellular integrity and prevent disease [4]. This process relies on the coordinated action of both the innate and adaptive immune system, with T cells playing a particularly central role. Through continuous surveillance, the immune system identifies and targets abnormal cells before they can become tumors. Immunosurveillance is essential for maintaining homeostasis and protection from cancer development and progression [4]. Once a tumor has managed to evade the immune system and become a cancerous mass, its characterization is essential for understanding tumor behavior and determining the clinical efficacy of treatments. Liquid tumors, such as leukemias and lymphomas, circulate through the bloodstream and primarily affect the blood, bone marrow (BM), and lymph nodes (LN). In contrast, solid tumors form localized masses in different organ systems. The response of these tumors to treatment is often categorized as ‘hot’ or ‘cold’ [5].

Hot tumors typically feature high levels of immune cell infiltration, particularly cytotoxic T lymphocytes (CTL) and T helper cells (T_H_), and a robust presence of antigen-presenting cells (APC) and chemokines like CC-chemokine-ligand 2 (CCL2), CCL3, CCL4, CCL5, CXC-ligand 9 (CXCL9), and CXCL10, which recruit and activate T cells, thereby creating a proinflammatory microenvironment. Consequently, they show a good response to treatment, typically characterized and supported by this high level of immune cell infiltration, which creates an immunologically active environment, making them responsive to potential immunotherapies [5,6,7,8]. These tumors also exhibit a high mutational burden, leading to the production of tumor-specific neoantigens that potentially enhance immune recognition. This robust immune activity results in highly tumor-responsive immunotherapeutic approaches, as the immune system is already primed to target the tumor [5,9,10].

In contrast, cold tumors show poor treatment response due to limited immune activity within the tumor immune microenvironment (TIME). They display a low immune cell infiltration, a lack of immune cell activity and of the tumor-specific neoantigen presentation necessary for effective anti-tumor responses. As a result, cold tumors generally show a poor response to immunotherapy, as the immune system is unable to efficiently recognize and attack the tumor due to the absence of immune cell engagement [10,11,12]. Several factors influence the efficacy of treatment, including the presence and activity of immune checkpoint molecules (IC), tumor-infiltrating leukocytes (TIL), tertiary lymphoid structures (TLS), microsatellite instability (MSI), and the tumor mutational burden (TMB), as well as the presence of neoantigens in particular. IC such as programmed cell death protein 1 (PD-1), programmed death-ligand 1 (PD-L1), cytotoxic T-lymphocyte associated protein 4 (CTLA-4), T cell immunoglobulin and mucin-domain containing-3 (TIM-3), and lymphocyte-activation gene 3 (LAG-3) are immune regulators that can trigger immune escape by promoting tumor progression and inhibiting effective immune responses [13,14]. The tumor-infiltrating leukocytes include both proinflammatory immune cells, such as dendritic cells (DC), cytotoxic CD8^+^ T cells (CTL), CD4^+^ T helper cells (T_H_), and natural killer cells (NK), as well as immunosuppressive immune cells like tumor-associated macrophages (TAM), regulatory T cells (T_REG_), myeloid derived suppressor cells (MDSC), and tumor-associated neutrophils (TAN), negatively regulating tumor immunity. TLS, structures at the tumor margin containing T cells and B cells, help to promote immune cell infiltration and enhance immune responses within solid tumors [15,16]. The presence of MSI, caused by mutations in mismatch repair (MMR) genes or expression abnormalities, often results in a poor immune response [17]. MSI in glioblastoma multiforme (GBM) occurs at a low frequency, with approximately 8.5% of cases exhibiting low-level MSI. This phenomenon is particularly relevant in recurrent tumors, where the accumulation of genetic alterations over time may contribute to tumor evolution. However, alterations in MMR, which are responsible for maintaining genomic stability, do not significantly correlate with tumor prognosis or survival outcomes in GBM patients. While MSI contributes to genomic instability within the tumor, it is generally considered more of a secondary factor rather than a primary driver of tumor initiation [18,19]. Conversely, a high TMB, which represents the number of nonsynonymous mutations in a tumor’s genome, may increase the likelihood of de novo antigen acquisition, also referred to as neoantigens, enhancing the potential for a better immune response and therefore checkpoint inhibitor (ICI) response. Specific gene mutations, such as those in tumor protein P53 (TP53), a tumor suppressor gene, or intracellular adhesion molecule 1 (ICAM-1), a cell adhesion molecule, can also affect tumor behavior and immune interactions. Tumor suppressor genes play a crucial role in maintaining cellular integrity by regulating processes such as cell cycle progression, DNA repair, and apoptosis. TP53, one of the best-known tumor suppressor genes, is particularly important in these processes. It functions as a key regulator of the cell cycle, initiating DNA repair mechanisms in response to damage, as well as promoting apoptosis when cellular damage is irreparable. These mutations impair the tumor’s ability to repair DNA damage and respond to chemotherapy-induced stress, contributing to the development of resistance [20,21,22]. Hence, it is of utmost importance to properly understand and characterize the tumor itself and its cellular and molecular features before deciding on the most promising therapy.

### 1.2. Glioblastoma Multiforme

A very prominent example of a cold tumor is the malignant brain tumor GBM, which is recognized as the most aggressive and lethal primary brain tumor in adults, representing a formidable challenge in neuro-oncology [23]. It belongs to a broader category of tumors known as gliomas, which arise from glial cells. These glial cells play a vital role in supporting and protecting neurons within the central nervous system. GBM accounts for approximately 15% of all primary brain tumors and constitutes 50% of all gliomas, making it a significant concern in terms of both incidence and impact [24]. Globally, its annual occurrence is estimated at 3.2 cases per 100,000 individuals. Despite advances in treatment modalities, including surgical resection, radiotherapy, and chemotherapy with alkylating temozolomide (TMZ), the prognosis for GBM remains dismal [25,26,27,28]. Median survival following diagnosis and comprehensive treatment ranges from 12 to 15 months [29], highlighting the importance of new therapeutic modalities. As per World Health Organization (WHO) classification, GBM is designated as a grade IV tumor, the highest level of malignancy [30]. This classification reflects its aggressive nature, characterized by rapid growth, extensive infiltration into surrounding brain tissue, and high-grade vascularization. The heterogeneity of GBM further complicates its management. The tumor has multiple genetic mutations and alterations, as well as a complex tumor microenvironment (TIME). This complex interplay of genetic and microenvironmental factors contributes to the failure of current treatment approaches to achieve long-term remission or cure. The relentless progression and poor outcomes associated with GBM underscore the critical need for continued research into its molecular mechanisms and the development of innovative treatment paradigms [28]. GBM is driven by a number of genetic alterations that contribute to its aggressive behavior and resistance to treatment.

With the development of new technologies and the introduction of molecular biomarkers in the diagnostic process, tumors of the central nervous system (CNS) are being re-grouped in the new WHO classification, improving their grading and denoting the biomarkers that have a prognostic impact [31]. For GBM, key genetic changes include mutations or dysregulations in the epidermal growth factor receptor (EGFR), TP53, phosphatase and tensin homolog (PTEN), telomerase reverse transcriptase (TERT), and O-6-methylguanine-DNA methyltransferase (MGMT). Whereas previously isocitrate dehydrogenase (IDH) mutant tumors were classified as grade IV gliomas, they are now classified as astrocytomas, and GBM as IDH wild-type [28,31].

As described above, mutations in TP53 are common in cold tumors and are associated with resistance to chemotherapy, particularly temozolomide (TMZ), the standard-of-care treatment for GBM [26], as the loss of function mutation in TP53 enables the tumor cells to evade apoptosis. Additionally, overexpression or mutation of EGFR enhances tumor cell proliferation, survival, and growth, while inactivation of PTEN removes constraints on tumor cell proliferation. IDH mutations, typically found in secondary GBM and nowadays not classified as GBM any longer, are associated with a better prognosis compared to the wild-type form seen in primary GBM [31,32,33]. Mutations in TERT facilitate continuous cellular division by preventing senescence. Only epigenetic silencing of MGMT enhances the tumor’s responsiveness to alkylating chemotherapeutic agents, such as TMZ. Together, these alterations enable uncontrolled cell proliferation, enhanced survival, and resistance to apoptosis [34]. In 2010, Verhaak et al. introduced a gene expression-based molecular classification that divided GBM into four distinct subtypes—proneural, neural, mesenchymal, and classical—each exhibiting unique molecular characteristics and clinical behaviors. The proneural subtype is marked by amplification of the platelet-derived growth factor receptor A (PDGFRA) gene and mutations in IDH1, and is associated with a better prognosis compared to the other subtypes [34,35,36]. The neural subtype is rare and presents a distinct gene expression profile, with high levels of nestin, and low expression of hexaribonucleotide binding protein-3 (NeuN) and oligodendrocyte-specific protein (OSP), indicating a biological behavior that differs significantly from the other subtypes [37,38]. The mesenchymal subtype, characterized by the loss of neurofibromin 1 (NF1) and mutations in TP53, is associated with increased invasion, angiogenesis, and a more aggressive clinical course, leading to a poorer prognosis. The classical subtype is defined by epidermal growth factor receptor (EGFR) amplification, loss of PTEN, and mutations in cyclin-dependent kinase inhibitor 2A (CDKN2A), all of which contribute to aggressive tumor behavior and a negative prognosis. Additionally, rare subtypes of GBM, such as gliosarcoma and giant cell glioblastoma, present unique pathological features and clinical challenges [36,39]. On top of the different molecular subtypes, intertumoral heterogeneity, characterized by distinct subpopulations of cancer cells, such as glioma stem cells, with different genetic and phenotypic profiles, further complicates treatment [40].

## 2. The Tumor Immune Microenvironment of GBM

### 2.1. Landscape of the TIME

One of the main reasons why GBM is referred to as a “cold” or “immunologically hostile tumor” is its TIME, which is highly dynamic and plays a critical role in tumor progression, immune evasion, and therapeutic resistance [12]. Characterized by invasive growth, angiogenesis, inflammation, hypoxia, necrosis, and immune suppression, the TIME fosters a favorable niche for tumor survival. Despite the low immunosurveillance and the immunologically cold environment, a hallmark of GBM is its strong infiltration by non-neoplastic cells, like immune cells, endothelial cells, and fibroblasts, whose cellular interaction fuels the pro-tumorigenic forces in the TIME with their anti-inflammatory cytokines, chemokines, growth factors, or direct cellular interaction [40,41,42]. The recruitment and expansion of anti-inflammatory immune cells leads to the formation of a highly immunosuppressive TIME and eventually to tumor immune evasion and progression [43].

Cancer-associated fibroblasts (CAF) are not only a major component of the TIME, but actively shape the TIME by promoting extracellular matrix remodeling and supporting invasive tumor phenotypes, thereby directly or indirectly influencing cancer cell behavior through their extensive interaction with cancer cells or stromal cells. It has long been a matter of debate as to whether CAF exist in GBM. However, newer studies have shown via single-cell RNASeq that GBM-associated fibroblasts exist and that they render the microenvironment more pro-tumorigenic by recruiting monocytes and promoting their differentiation and polarization into M2 macrophages by signaling through Toll-like receptor 4 (TLR4) and thereby contributing indirectly to the immunosuppressive character of the TIME [44,45,46,47]. In contrast to proinflammatory M1 macrophages that help fighting the tumor, M2 macrophages secrete anti-inflammatory cytokines, which contribute to the influx of further immunosuppressive cells and thereby to tumor progression, which will be described in more detail later in this paper.

In addition to the stromal cell compartment, the immunosuppressive TIME of GBM is largely determined by its immune cells. While the brain has long been considered an immune-privileged organ due to the presence of

an intact blood–brain barrier (BBB) restricting the immune cell entry,the absence of typical lymphatic drainage, thereby preventing the trafficking of antigens and DC to lymph nodes [48],rareness of APC [49], anddecreased expression of MHC in brain tissue that limits immune response [49],

more recent studies have suggested a fully functional innate and adaptive immune system within the brain tissue by showing that the CNS has functional lymphatic vessels lining the cerebral sinuses. These structures have all the molecular features of lymphoid endothelial cells and are capable of shedding macromolecules and immune cells from the meninges and cerebrospinal fluid (CSF) [50,51,52]. They are also associated with cervical lymph nodes, which are the main site of systemic activation of CNS-specific T cells [53] induced by antigens and possibly dendritic cells (DC) [54]. Professional antigen-presenting cells (APC), like macrophages, DC, and B cells, have been detected in many regions of the brain, including the leptomeninges, the ventricles (choroid plexus), and the perivascular spaces [55].

Additionally, CNS-resident microglia, the brain’s resident immune cells, normally respond to neuronal damage and eliminate the impaired cells by phagocytosis [56]. Microglia interact with tumor cells through glutamatergic signaling and thereby enhance tumor proliferation [57]. They adopt an anti-inflammatory phenotype in GBM, contributing to the immunosuppressive tumor microenvironment. Through crosstalk with glioblastoma stem cells (GSC), microglia enhance tumor malignancy by supporting GSC self-renewal and the tumor-initiating capacity, irrespective of environmental constraints. This interaction reinforces the invasive and resistant nature of glioblastoma. Moreover, microglia influence the recruitment and function of regulatory T cells (T_REG_), further shaping an immunosuppressive milieu that hinders effective anti-tumor immunity [57,58,59].

Besides microglia, there is an even bigger population of monocytic cells in the GBM TIME, which mainly consists of tissue resident cells rather than circulating ones. The tumor-associated macrophages (TAM), professional APC specialized in the detection, phagocytosis, and elimination of pathogens and cell debris, make up around 30–50% of the tumor mass [60,61]. The recognition of pathogens is mediated by the pattern recognition receptors (PRR), which are therefore also used as surface markers for their identification. For example, all TAM subtypes, the pro-inflammatory M1 and the immunosuppressive M2 macrophages, express the mannose receptor CD206, while only the M2 subtype is positive for the scavenger receptor CD163 [62]. It is now known that TAM originate from circulating monocytes from the bone marrow and accumulate around and in the tumor during the first steps of tumor development and are activated by the secretion of chemokines and cytokines such as macrophage-colony stimulating factor (M-CSF), granulocyte-macrophage colony-stimulating factor (GM-CSF), and CCL2 [63,64,65].

Increasing tumor size, the formation of an intra-tumoral vascular network, as well as the highly hypoxic microenvironment of GBM have been shown to skew the highly plastic CNS-macrophages towards the M2-TAM phenotype by an activation of the STAT3 pathway [66]. Moreover, further vascularization leads to a stronger infiltration of CCR2^+^ monocytes in response to the secretion of several chemokines, such as CCL2, CCL18, CCL20, CXCL12, M-CSF, GM-CSF, and VEGFA [61,62,63,65,66], and the differentiation of the infiltrated cells into immunosuppressive M2-like TAMs. They contribute to the production of pro-tumorigenic factors such as IL-10 and IDO [29] and facilitate tissue remodeling and cell invasion through matrix metalloproteinases (MMP) [67,68,69,70]. By blocking CSF-1 and macrophage inhibitory cytokine-1 (MIC-1), TAM are transformed into the M2-phenotype, which subsequently loses its phagocytic capacity, suppresses CTL proliferation, and enhances the effect of T_REG_.

The presence of T_REG_ in GBM is associated with a poor prognosis due to these immunosuppressive properties [70,71]. They are recruited by GBM-derived CCL22 and CCL2, which attract T_REGs_ to the tumor region via CCR4 [72]. As an abundant population in the TIME of GBM, they contribute significantly to its immunosuppressive character and are characterized by a CD3^+^/CD4^+^/CD25^high^/CD127^−/low^/FoxP3^high^/CD45RA^−^/CD45R0^+^/CCR7^+^/CD62L^−^/CTLA4^+^ immunophenotype. The immunosuppressive nature of T_REG_ is a consequence of several mechanisms. For one, they inhibit the production of IL-2 and IFN-γ [73], thereby switching the immune response from a tumor-directed cytotoxic T_H_1-mediated immunity to a T_H_2-mediated response [74,75]. In addition, IL-2 deprivation leads to the disruption of metabolic pathways in effector T cells, driving them towards apoptosis [73]. They are also high producers of IL-10, TGF-ß, and IL-35, cytokines which inhibit effector T-cell proliferation and induce the (DC) maturation towards a tolerogenic phenotype, which in turn induces additional T_REG_ formation [70,74,75,76]. Another mechanism of immunosuppression is the expression of immune checkpoint (IC) molecules on GBM-associated T_REGs_, including programmed cell death protein 1 (PD-1), whose ligand PD-L1 is expressed by GBM cells as well as microglia and TAM, and their interaction promotes T_REG_ activity and consequently the inhibition of the proliferation and function of effector T_H_ cells and CTL [77,78,79]. CTLA-4, another immune checkpoint molecule expressed on T_REGs_ and activated T_H_ cells, appears to be critical for immune evasion in GBM [80].

Despite the urgent need and presence of hostile neoplastic cells, effector T cells in the GBM TIME are only present in low numbers and are largely dysfunctional. Moreover, chronic antigen exposure drives CD8^+^ CTL into an exhausted state, marked by upregulation of exhaustion markers such as T-cell immunoreceptors with immunoglobulin and ITIM domains (TIGIT), LAG-3, TIM-3, PD-1 or CTLA-4 [81,82] and their suppression by T_REG_. GBM also activates pericytes, which adopt an immunosuppressive phenotype, further suppressing T-cell activation. The expression of immune checkpoint molecules within the TIME supports this immunosuppressive state [83]. For instance, GARP (glycoprotein A repetitions predominant), expressed on GBM cells, inhibits the effector function of T cells. This immune exhaustion profile significantly limits the efficacy of immune checkpoint inhibition (ICI), and therefore reinvigoration of exhausted T cells remains challenging in GBM [84].

In addition to an exhausted or dysfunctional/tolerogenic phenotype of effector T cells in the GBM TIME, CD4^+^ T_H_ are often skewed toward a T_H_2 phenotype, which is less effective in anti-tumor immunity compared to T_H_1. In GBM, tumor-driven immunomodulatory effects create a T_H_2-biased immune environment that suppresses effective anti-tumor immunity. This shift is evident in the peripheral blood, where elevated levels of IL-4 and IL-13 promote T_H_2 differentiation. The presence of M2-like macrophages sustains the T_H_2-biased milieu. Additionally, the TIME secretes a variety of soluble factors and exosomes that support and amplify T_H_2 responses. This predominance of T_H_2 cells within GBM undermines cytotoxic immune responses, contributing to immune evasion and tumor progression [85].

B cells, although relatively rare, may contribute by producing antibodies against tumor antigens, although their importance so far remains underexplored. B cells play a dual role in immunomodulation of GBM, particularly through their regulatory subset B_REG_, which suppresses CD8^+^ T cell activation. This suppression is mediated by the expression of inhibitory molecules such as PD-L1 and the secretion of immunosuppressive cytokines, such as IL-10 and TGF-β. MDSC further enhance B_REG_ differentiation by transferring PD-L1-bound vesicles to B cells, thereby enhancing their immunosuppressive capabilities. The presence or absence of B cells significantly alters the immune landscape in GBM: their absence correlates with increased M2 macrophages, while their presence is associated with a reduction in CD8^+^ T cells, weakening anti-tumor immunity. TIM-1^+^ B cells have emerged as a potential regulator of the balance between immune activation and suppression, as TIM-1 expression correlates with other inhibitory molecules that affect the TIME. These findings suggest that B cells, particularly B_REG_ and TIM-1^+^ subsets, are critical modulators of GBM immunity and potential targets for therapeutic intervention [86,87,88].

In addition to conventional immune populations, specialized immune cells such as γδ T cells and eosinophils play an emerging role in the GBM TIME. γδ T cells, a small subset of T cells located primarily in tissues such as the thymus, lymphoid organs, and epithelial layers are characterized by their distinct T-cell receptor (TCR) chains, Vδ1, and Vγ9Vδ2. The major subset, Vγ9Vδ2 T cells, exhibits potent anti-tumor activity when activated, directly lysing tumor cells and secreting pro-inflammatory cytokines such as interferon gamma (IFNγ) and tumor necrosis factor alpha (TNFα) [89]. Unlike conventional T cells, γδ T cells recognize stress-induced antigens such as MHC class I chain-related sequence A (MICA) and MICB on tumor cells, as well as natural killer group 2 member D (NKG2D) ligands, without the restriction of human leukocyte antigen (HLA) molecules. Notably, therapeutic agents such as TMZ and poly-ADP ribose polymerase (PARP) inhibitors can enhance the expression of these ligands, potentially amplifying the cytotoxic effects of γδ T cells [90,91].

Eosinophils, traditionally associated with allergic responses, are now recognized as modulators of the TIME with potential roles in anti-tumor immunity. They can influence the infiltration of immune cells, including T cells, and are linked to a reduction in immunosuppressive M2 macrophages within the TIME. This shift correlates with a transition from an immunosuppressive environment to a more immunopermissive state, which could enhance the efficacy of immune responses against GBM [92]. They release granulocyte-macrophage colony-stimulating factor (GM-CSF), which establishes a paracrine loop with GBM cells, as these cells also secrete GM-CSF. This cytokine enhances eosinophil survival, activation, and the production of growth factors, sustaining their activity within the tumor microenvironment. Additionally, eosinophils secrete pro-inflammatory cytokines such as IL-5 and eotaxin, which are associated with increased infiltration of effector T cells and a reduction in immunosuppressive M2 macrophages in GBM. These effects contribute to a shift toward a more active anti-tumor immune response. Moreover, eosinophils are linked to elevated levels of CTL, further strengthening their role in enhancing anti-tumor immunity [93,94].

In addition to γδ T cells and eosinophils, neutrophils are also key immune players in the GBM TIME. These cells significantly influence tumor progression and immune modulation, often exhibiting immunosuppressive functions, particularly within the hypoxic regions of the tumor. In these areas, neutrophils contribute to tumor growth and immune evasion by undergoing reprogramming, which alters their typical immune defense roles (see Figure 1).

The functional diversity of neutrophils is reflected in their classification into tumor-associated neutrophils (TAN) and polymorphonuclear myeloid-derived suppressor cells (PMN-MDSC), with each subtype playing distinct roles in the TIME [95,96,97]. While neutrophils can promote tumor progression by inhibiting T-cell activation and enhancing tumor cell survival, they can also exert anti-tumor effects. A key mechanism is the neutrophils’ expression of PD-L1, which interacts with PD-1 on T cells, inducing exhaustion and reducing cytotoxic activity. In addition, neutrophils produce reactive oxygen species (ROS) that interfere with T-cell receptor (TCR) signaling, promoting a state of unresponsiveness and impaired activation. Another mechanism is trogocytosis, a process through which neutrophils adhere to T cells and extract membrane fragments, disrupting critical signaling pathways required for effective T-cell activation and function. Together, these mechanisms undermine T-cell-mediated anti-tumor immunity [98,99,100]. However, through the production of ROS and proinflammatory cytokines, neutrophils can also directly target and kill tumor cells. Furthermore, neutrophil interactions with other immune cells, such as macrophages and T cells, significantly influence the overall immune landscape within the TIME. Depending on the context, these interactions can lead to either tumor promotion or immune suppression, highlighting the dual role that neutrophils can play in both supporting and opposing tumor growth. The complex balance between these opposing effects underscores the importance of neutrophils as critical regulators of the immune response in GBM and their potential as targets for future therapeutic strategies. In addition to detailed characterization of the cells, current studies on neutrophils are using both the number and subdivision of neutrophils as indicators to better categorize the success of therapy and the course and severity of the disease [101,102,103,104].

Together, these less conventional immune cell populations contribute to the complex interplay of factors within the TIME, offering novel avenues for therapeutic exploration in GBM. These features of the TIME create a permissive and highly resistant environment that facilitates tumor growth and undermines effective immune responses (see Figure 1).

### 2.2. Immune Cell Interactions

Immune cell–molecule interactions within the GBM TIME are complex and contribute significantly to tumor progression and immune evasion. TAM are notably influenced by glioblastoma-derived exosomes (GDE), which are critical modulators of the tumor microenvironment, significantly contributing to immune suppression and tumor progression [105,106]. GDE are either present within the tumor or recruited by M1 macrophages, which are in turn transformed into the immunosuppressive M2 phenotype. This transformation is mediated through diverse mechanisms like the upregulation of PD-L1 [105]. Additionally, extracellular vesicles secreted by GSC play a role in altering the metabolism of DC by inducing lipid accumulation and ferroptosis via the nuclear factor erythroid 2-related factor 2 (NRF2)/glutathione peroxidase 4 (GPX4) pathway, impairing their function [107]. GDE also contain components of the signal transducer and activator of transcription 3 (STAT3) signaling pathway, which is strongly associated with immune suppression and enhanced tumor growth [92,94]. These exosomes carry arginase-1, a key immunosuppressive protein, and ICAM-1, which facilitates macrophage migration, proliferation, and phagocytosis, ultimately supporting tumorigenesis, particularly under hypoxic conditions [108]. Additionally, polysialic acid in GDE interacts with sialic acid-binging Ig-like lectin 16 (Siglec-16) on TAM, promoting a pro-inflammatory response, while branched-chain ketoacids (BCKA) secreted by GBM cells are taken up by TAM, reducing their phagocytic activity. Similarly, microglia, the resident immune cells of the central nervous system, are also reprogrammed in the microenvironment of GBM.

Microglia are affected by lysophosphatidic acid (LPA) secreted by GBM cells, thereby activating them through the lysophosphatidic acid receptor 1 (LPA1), which in turn promote tumor proliferation and migration [109]. Further, IL-6 and IL-8 secreted by microglia induce GBM cell proliferation and invasiveness, while adenosine signaling through A1 adenosine receptors further modulates the microglial response to GBM. Other factors such as TGF-α, EGF, and CSF-1 support microglial activation and their survival, thus enhancing GBM cell migration and invasion [110]. CSF-1 was investigated as a potential target molecule for the shift of the immunosuppressive M1 macrophages into the proinflammatory M1 subtype, but its blockage in the context of such a TIME modulation showed little to no success [111].

A key factor by which the GBM cells contribute actively to shape the immunosuppressive TIME and thereby perpetuate their own growth is the active production of the CC-chemokine ligand 2 (CCL2), the predominant chemokine that recruits T_REG_ and MDSC, the key population for immune evasion of the GBM cells [72,76]. Moreover, CCL2 is also produced by macrophages and microglia, and their elevated levels correlate with poor patient survival. Factors such as M-CSF and IL-34 promote the differentiation of monocytic MDSC (M-MDSC), which enhance immunosuppressive functions against CTL. CXCL1 and CXCL2 facilitate MDSC migration to the tumor contributing to the disruption of CTL accumulation. Additionally, Siglec receptors on MDSC transmit inhibitory signals, enhancing their suppressive phenotype. To understand the effects of MDSC in the TIME, a distinction must be made between granulocytic-MDSC [112] (G-MDSC) and monocytic-MDSC (M-MDSC). While the immunosuppressive effect of G-MDSC is mainly mediated by the production of arginase-1, ROS, and prostaglandin E_2_ (PGE_2_), M-MDSC express arginase-1 and PD-L1 and secrete TGF-β and IL-10 [112]. TGF-β, IL-10, and PD-L1 are immunosuppressive cytokines released and/or expressed by GBM cells enhancing the inhibition of T-cell activation and promoting immune evasion [113,114]. Through the signaling of these molecules, effector T cells in the TIME are often rendered dysfunctional by upregulating immune checkpoint molecules such as PD-1 and TIM-3, leading to T-cell exhaustion and loss of their cytotoxic potential [115]. Additionally, the ligands of these immune checkpoint molecules, such as Galectin-3 and PD-L1, can induce T-cell apoptosis and support the differentiation of T_REG_, further suppressing effective anti-tumor immunity [116,117].

TGF-β plays a pivotal role in GBM pathophysiology by mediating tumor progression and immune evasion. Produced by GBM cells as well as macrophages and microglia within the TIME, TGF-β also promotes the differentiation of T_REG_. Moreover, TGF-β impairs DC maturation and their antigen-presenting ability, thereby limiting the stimulation of tumor-specific effector T cells [118]. It is also associated with the downregulation of activating receptors such as NKG2D, further impairing immunosurveillance. TGF-β can establish positive feedback loops with IL-33, a cytokine secreted by tumor-initiating cells, amplifying its immunosuppressive effects. Furthermore, TGF-β interacts extensively with CAF and stromal cells, shaping the tumor microenvironment to favor immune escape and disease progression. By driving the transition of cells to a myofibroblast phenotype, TGF-β facilitates extracellular matrix (ECM) remodeling, which enhances structural support for tumor growth and invasion [119,120,121,122].

Similar to TGF-β, lactate dehydrogenase 5 (LDH5) secreted by GBM cells induces the expression of NKG2D ligands on myeloid cells, which in turn downregulate NKG2D receptors on NK cells, impairing their ability to recognize and kill tumor cells [123]. However, IL-15 can enhance NK-cell proliferation and cytotoxicity, while proliferating cell nuclear antigen (PCNA) overexpression in GBM may facilitate NK cell targeting of tumor cells [124].

Finally, B cells in GBM are influenced by FMS-like tyrosine kinase 3 ligand (Flt3L), which enhances B-cell activity and supports T-cell-mediated anti-tumor immunity. However, upregulation of PD-1 and TIM-3 in GBM can inhibit B-cell functions, further contributing to the immune suppression within the TIME [125].

Of note, Flt3L is also used as a maturation factor for manufacturing in vitro plasmacytoid DC (pDC), which have gained substantial interest in the use of cellular immunotherapies as they are associated with the stimulation of strong effector T-cell responses [126]. These intricate immune cell–molecule interactions highlight the multifaceted ways in which the GBM TIME suppresses anti-tumor immune responses and supports tumor progression.

Potential therapeutic targets include inhibitors of T_REG_ function or MDSC activity, as well as agents for shifting the M2-suppressive TAM into pro-inflammatory M1 macrophages to disrupt the immunosuppressive axis, or monoclonal antibodies that can revive the dysfunctional state of effector T cells. Moreover, biomarkers associated with exhaustion, such as exhaustion-related gene expression profiles, are being explored to stratify patients and predict their responses to immunotherapy, distinguishing responders from non-responders. The overarching goal in GBM treatment is to either overcome the immunosuppressive tumor environment or develop combinatorial approaches that reinvigorate exhausted T cells and thereby enhancing their anti-tumor activity [127,128]. These strategies hold promise for overcoming the immunosuppressive barriers imposed by GBM, offering a pathway to more effective immunotherapeutic interventions.

### 2.3. Therapies Targeting the Immunosuppressive TIME in GBM

#### 2.3.1. Standard of Care

The management of GBM typically involves a combination of therapies aimed at reducing tumor size, improving symptoms, and extending survival. Surgical resection is often the first step, aiming to remove as much of the tumor as possible, although complete removal is challenging due to the infiltrative nature of GBM [129,130]. Typically, and as the standard of care treatment for GBM, radiotherapy is administered post-surgery to destroy any remaining cancer cells and shrink the tumor. It works in tandem with TMZ, an alkylating chemotherapy agent that interferes with DNA replication, enhancing the effects of radiotherapy and reducing the likelihood of tumor recurrence [27,131]. In addition to these traditional treatments for GBM, recent improvements are pushing the boundaries of therapeutic strategies, offering promising avenues for improved patient outcomes. For example, proton radiation therapy, when combined with seco-duocarmycin SA (sDSA), a cytotoxic drug, demonstrates enhanced efficacy in killing GBM cells, with synergistic effects that improve treatment outcomes or even the use of combinatory chemotherapies for some subgroups of GBM [132,133].

#### 2.3.2. Tumor Treating Fields (TTF)

A rather different and relatively new approach for GBM treatment involves tumor-treating fields (TTF). TTF are low intensity (1–3 V/cm), intermediate frequency (200 kHz) alternating electric fields, which are applied to the patient’s head and therefore the tumor [134,135]. It is suggested that TTF therapy has an immunostimulatory effect and could help in overcoming the immunosuppressive TIME by [136]:Disrupting the polymerization of tubulin and localization of septin, molecules in the nucleus of dividing cells. TTF impairs the assembly of the mitotic spindle structure that is essential for chromosome segregation and cytokinesis. These interferences lead to cell cycle arrest, chromosome mis-segregation and breakage, and consequently, TP53 dependent or independent apoptosis of the tumor cells [137,138,139].Interfering with DNA repair mechanisms by downregulating breast cancer (BRCA) 1 and 2 genes and Fanconi anemia pathway genes, important molecules for DNA damage and repair mechanisms [138,140].Altering the structure of the cell membrane of GBM cells, making them more permeable for distinct substances and therefore more susceptible to potential chemotherapeutic agents [141].Making the blood–brain barrier more permeable by disrupting its tight junctions [103].Inducing immunogenic cell death and endoplasmic reticulum (ER) stress, such as the translocation of calreticulin to the cell surface, the release of high mobility group B1 protein (HMGB1), and the presence of extracellular ATP.Improving the phagocytosis of TTF-treated cells by DC [142].

Through all these different modes of action, TTF therapy has been shown to be effective in the treatment of GBM. Patients receiving TTF in combination with TMZ chemotherapy have shown longer progression-free survival and overall survival than patients receiving TMZ alone [27,143,144]. As a result, TTF is slowly becoming part of the standard of care for GBM. However, TTF has limited survival benefit, is costly, and its use affects patients’ quality of life due to its inconvenience. In order to overcome the long-term wear and tear of the TTF device and still extend the survival benefit for the patient, combinatorial approaches are being explored. As TTF is able to induce immunogenic cell death, a potential combination with immunotherapies is being considered. Recent studies have investigated the combination of TTF with the immune checkpoint inhibitor anti-PD1, which resulted in tumor shrinkage and increased infiltration of IFNγ^+^ T cells in mouse models [142]. In contrast, Diamant et. al. showed that application of TTF reduced T-cell proliferation and the viability of actively proliferating but not resting T cells without affecting their functionality [145]. Therefore, it would be possible to use TTF first and then possibly apply cellular immunotherapies. However, further research is needed to explore these options in more detail.

#### 2.3.3. Treatment with Small and Large Molecules

Despite these physical treatment options, the prognosis remains poor, leading to the exploration of additional strategies such as treatment with small molecules or monoclonal antibodies against certain chemokines.

One of the most studied therapeutic interventions is anti-angiogenic therapy, such as Bevacizumab, which targets the formation of blood vessels within the tumor, thereby restricting the tumor’s nutrient supply and inhibiting its growth by blocking VEGF, a chemokine responsible for angiogenesis. One of the hallmarks of GBM is heavy vascularization with blood vessels that have grown too quickly and are mostly dysfunctional, leading to necrotic tumor cores that perpetuate the hypoxic state of the GBM TIME and thereby contribute to its aggressiveness. Blocking VEGF could therefore interrupt this pro-tumorigenic process [146,147].

More targeted therapies focus on inhibiting specific molecular pathways within the tumor cells themselves. For example, phosphatidylinositol 3-kinase (PI3K)/protein kinase B (Akt) acts as a signaling enzyme for cell proliferation and intracellular trafficking in GBM, and the enhancer of zeste homolog 2 (EZH2) acts as a histone methylator that causes transcriptional repression to inhibit tumor suppressor genes in many cancers. Together, these molecules are essential for cancer cell division and proliferation, which drive tumor progression and resistance by stabilizing the tumor proliferation machinery. Their direct inhibition could therefore have an effect on dividing and proliferating GBM cells [148].

Another approach would be to target the differentiation process of GBM cells through kinases with so-called differentiation therapies. One of the best-known targets for such therapies is the PDGF/PDGFR pathway. It has been described that *PDGFRA* gain/amplification is a predictor of poor prognosis in IDH wild-type GBM [149]. Lane et al. and others have shown that inhibition of PDGF-Rα/β in vitro induces outgrowth of neurite-like processes in GBM cell lines and GSC, suggesting differentiation into neural-like cells, while reducing proliferation and invasion by upregulating the phosphatase dual-specificity phosphatase 1 (DUSP1) and downregulating phosphorylated p38 mitogen-activated protein kinase (MAPK) [150]. Pandey et al. have reviewed the PDGF mechanisms in more detail and summarized clinical trials that have already inhibited this pathway, e.g., by Crenolanib, in several tumor entities [151].

Targeting the cells of the GBM TIME may have an indirect but equally effective consequence on tumor inhibition. In particular, the immunosuppressive properties of tumor-associated macrophages have become the focus of therapeutic intervention, with the hope of modulating the TIME into a more immunopermissive environment by blocking or targeting TAM. As mentioned above, targeting the CSF1/CSF1R axis via various approaches, including tyrosine kinase inhibitors (Pexidartinib) or the use of monoclonal antibodies (Emactuzumab, Cabiralizumab), is currently being investigated in ongoing clinical trials [152].

The future of treatments with large molecules, such as monoclonal antibodies, is being sought through the development and use of bispecific antibodies (BsAbs), which can offer several advantages in the treatment of solid tumors, including improved immune infiltration, but still harbor a lot of challenges. While BsAbs can target the TIME and the tumor cells simultaneously by binding to TAA on cancer cells combined with other molecules expressed on immune cells (e.g., the CD3 receptor on T cells to activate them), they are still far from being the absolute solution. The tumor heterogeneity and lack of specific neoantigens in GBM make it very challenging to develop BsAbs targeting the majority of the tumor cells [153,154].

#### 2.3.4. Treatment with Immune Checkpoint Inhibitors

Using immunotherapies, especially the inhibition of immune checkpoint molecules (ICI), aims to boost the body’s immune response against GBM. This includes approaches like monoclonal antibodies against IC, anti-tumor vaccines and gene therapy, and CAR-T cells and other cellular therapies, designed to activate the immune system to specifically target and destroy tumor cells. For ICI, a wide range of different agents has been explored in the past decade with more or less promising outcomes [155]. In GBM, PD-1/PD-L1 is the best-characterized immune checkpoint pathway. In the majority of tumors, tumor cells [156] express PD-L1, although expression may be restricted to a minor subpopulation of tumor cells only (0–87%; median 2.8% [70]). However, it is also expressed by cells of the TIME, such as TAM and MDSC (reviewed in [70]). The respective receptor, PD-1, is expressed on CD4^+^ and CD8^+^ tumor-infiltrating T-cells (TIL) [156]. Expression on TIL is higher than on their counterparts in blood [157,158], and these cells are in a tolerized or dysfunctional state [11,12,157,159]. As previously described, besides PD-1, expression of TIM-3, LAG-3, and CTLA-4 has also been observed on infiltrating T cells in GBM [128]. Thus, the immune checkpoint pathways appear to play a central role in GBM immune escape and may contribute to preventing efficacy of other therapies. This tolerized or exhausted state of the T cells is not permanent, but appears to have to be maintained by receptor–ligand interactions. When the respective immune checkpoint receptors are blocked, e.g., by receptor or ligand-specific monoclonal antibodies, dysfunctional T cells can be reinvigorated: their function and proliferation is restored [128,160]. In many tumor entities, the application of monoclonal antibodies blocking immune checkpoint receptors or their ligands has resulted in a survival benefit for the patients [160]. However, in GBM, ICI has not shown any beneficial effect in several clinical trials [161,162,163,164,165,166,167,168,169]. One of the reasons might be that ICI can only enhance but not induce anti-tumoral responses. The low mutational load [170] and the low frequency of pre-existing anti-tumoral T-cell responses in GBM patients [171,172] may therefore limit the efficacy of ICI. In agreement with this, PD-1 or PD-L1 blockage combined with a dendritic cell vaccination (DCV) in mouse models resulted in CD8^+^ T-cell dependent long-term survival, which was not observed with the respective monotherapies [163,173]. Similar results have been obtained by Wang et al., who vaccinated GBM patients with personalized tumor-associated antigens (TAA)-pulsed DC combined with low-dose cyclophosphamide, poly I:C, Imiquimod, and anti-PD-1 antibody [174]. Thus, the immune checkpoint pathways appear to play a central role in GBM immune escape and may contribute to preventing different treatments, but could also be effective in combination with other therapies, like DCV.

#### 2.3.5. Treatment with Dendritic Cell Vaccination (DCV)

GBM is characterized by a weak natural anti-tumoral immune response, because of the poor immunogenicity and/or an immunosuppressive TIME (for review, see [70]). To overcome this immunosuppressive TIME, immunotherapeutic approaches such as DCV aim to activate and boost the anti-tumoral immune response in GBM. DCV is a type of active immunotherapy seeking to exploit the crucial role of DC in the initiation of T-cell responses. Patients are vaccinated with TAA-loaded DC to initiate a TAA-specific T-cell response that kills the tumor cells specifically and may prevent tumor recurrence through immunological memory [175]. Active immunotherapy of GBM with DCV has been pioneered by Liau et al., who described the vaccination of a GBM patient with recurrent disease with DC pulsed with eluted peptides of an HLA class I-matched GBM cell culture in 2000 [70,176,177]. Since then, numerous studies have been published (for a recent summary, see [70]), including six controlled, randomized trials [171,176,178,179,180,181]. Most patients undergo cytoreductive surgery prior to DCV, and an association between survival and the extent of resection has been reported [182]. Indeed, a state of minimal residual disease has been indicated to be beneficial for vaccination therapy [177,183]. Induction of antigen-specific immune responses has been observed in the course of DCV, whereby the detection of IFNγ production by certain cell types has been the most informative [171,172,174,184,185,186,187], but anti-tumoral cytotoxic responses [172,188,189] and an increase in tetramer positive cytotoxic T cells have been reported as well [172,188,189]. Several studies have identified immunological responders based on antigenic-target directed delayed-type hypersensitivity reactions, IFNγ responses, or cytotoxic responses, which increased in the course of vaccination and resulted in reported longer survival times for responders [172,177,181,185,186]. For newly diagnosed patients, median overall survival (mOS) ranged from 15 to 41.4 months, and the progression-free survival (PFS) ranged from 6 to 25.3 months. Meanwhile, several controlled studies have been published, six of them randomized [176,178,179,180,181].

Taken together, from the controlled clinical trials, it is still difficult to draw a robust conclusion as to the efficacy of DCV in GBM. Although DCV can elicit antigen-specific T-cell responses in a subset of patients, and survival appears to be improved, particularly in immunological responders, clinical efficacy appears to be limited and restricted to a subset of patients. Moreover, more recent studies have described a more complex immune circuit indicating that monotherapies, even DCV, show limited efficacy, and that combinatory approaches are mandatory [11,12,190].

#### 2.3.6. Treatment with CAR-T Cells

Another cutting-edge approach is chimeric antigen receptor (CAR) T-cell therapy, which has been extensively studied and is now in clinical use for the treatment of GBM. CAR-T therapy offers several advantages in GBM, in particular its ability to enhance anti-tumor efficacy through positive manipulation of T-cell trafficking. This strategy can be further enhanced when combined with other therapies, such as anti-VEGF agents, to counteract the immunosuppressive TIME. In addition, GBM-specific antigens, including EGFRvIII and IL-13Rα2, can be targeted to generate highly specific and potent anti-tumor responses. However, this approach faces significant challenges. The efficacy of the genetically modified CAR-T cells is dependent on the recognition of its specific antigen. However, tumor heterogeneity often leads to antigen loss and escape mechanisms, reducing long-term efficacy [191,192]. Furthermore, autologous CAR-T therapy is often constrained by the dysfunctionality and pre-existing exhaustion of patient-derived T cells. A notable risk is cytokine release syndrome (CRS), characterized by severe inflammatory cytokine storms, which poses substantial clinical management difficulties. Moreover, CAR-T cell exhaustion, particularly in the highly immunosuppressive GBM microenvironment, can lead to loss of cell memory function, further compromising therapeutic durability and effectiveness [193,194,195]. Despite these challenges, early preclinical and clinical results show that CAR-T cells targeting specific tumor antigens offer a promising way to overcome immune evasion and attack GBM cells more effectively.

Targeted drug delivery systems enabled by nanotechnology are revolutionizing GBM therapy by facilitating the crossing of the BBB, thereby improving the precision and efficacy of drug delivery [196]. Innovative strategies, such as the use of swimming nanorobots, are being developed to deliver therapeutic agents directly to tumor sites, potentially improving outcomes for patients with GBM [196]. Additionally, the sequential inhibition of PARP and bromodomain and extraterminal (BET) proteins has shown promise in enhancing anti-tumor effects while minimizing toxicity, presenting a new avenue for effective GBM treatment [197]. Moreover, RNA-based gene therapies, utilizing non-viral nanocarriers for drug delivery, aim to overcome treatment resistance and target stem-like tumor cells, offering a strategy to combat the recurrence of GBM [196]. To refine these approaches, the use of patient-derived xenograft (PDX) models, combined with patient-specific data and whole exome sequencing (WES), is advancing preclinical studies and enhancing the predictive relevance of therapies for individual patients [198]. These innovative treatment modalities reflect a significant shift towards more personalized, targeted approaches in GBM therapy, addressing the unique challenges posed by the tumor’s aggressive and heterogeneous nature (Table 1; Figure 2).

## 3. Conclusions and Perspectives

Despite its rarity, GBM is one of the most aggressive tumor entities worldwide and the most aggressive brain tumor. Most attempts to target it therapeutically have failed in the past. However, several steps have brought us closer to tackling glioblastoma, a “hard to hit target”. On the one hand, a better understanding of the molecular and phenotypic characteristics of the cancer has led to the development of several small and large molecules that directly target the tumor cells. On the other hand, improvements in technical approaches to address these issues, such as the development of spatial transcriptomics techniques, have contributed to a better understanding of the complex structure of the immunosuppressive TIME. In particular, the advanced understanding of the exhausted phenotype of anti-tumor immune cells has led to the development of immune checkpoint therapies and treatment with cellular therapies such as DCV and CAR-T cells.

We have reviewed most of the treatment options (Table 1; Figure 2), and although most have not shown clinical benefit to date, several promising clinical trials are underway. We believe that the approach of targeting the tumor and its TIME simultaneously, e.g., with bi- to tri-specific CAR-T cells [153,154], is likely to become the cornerstone of the next generation of cancer immunotherapy and bring new hope to patients with multi-resistant tumors such as GBM.

## Figures and Tables

**Figure 1 cancers-17-00817-f001:**
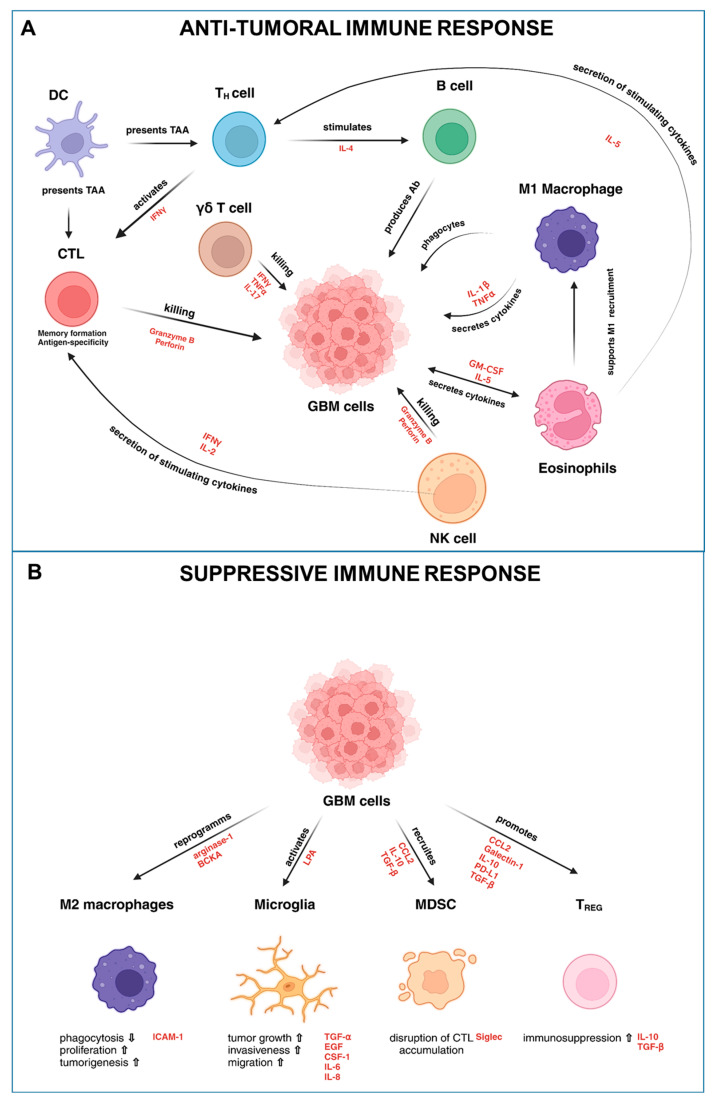
Immune cell interactions within the tumor microenvironment (TME) of glioblastoma multiforme (GBM). Tumor-infiltrating leukocytes (TIL) can either act (**A**) anti-tumoral and co-stimulate each other to attack the tumor mass, or they can act as (**B**) immunosuppressive regulators that are actively manipulated by GBM cells, or the TIL can manipulate each other to evade, reduce, or completely disable an active anti-tumoral immune response. DC: dendritic cells; TAA: tumor-associated antigens; TNFα: tumor-necrosis factor; GM-CSF: granulocyte-macrophage colony-stimulating factor; BCKA: branched-chain ketoacids; ICAM-1: intracellular adhesion molecule-1; LPA: Lysophosphatidic acid; MDSC: myeloid derived suppressor cells; CTL: cytotoxic T cells; IFN: Interferon; MDSC: myeloid derived suppressor cells; TREG: regulatory T cells; TGF-β: tumor growth factor beta; EGF: epidermal growth factor; CSF-1: colony-stimulating factor-1; CCL2: chemokine ligand 2; PD-L1: programmed cell death ligand-1. Created with BioRender.com (accessed on 24th January 2025).

**Figure 2 cancers-17-00817-f002:**
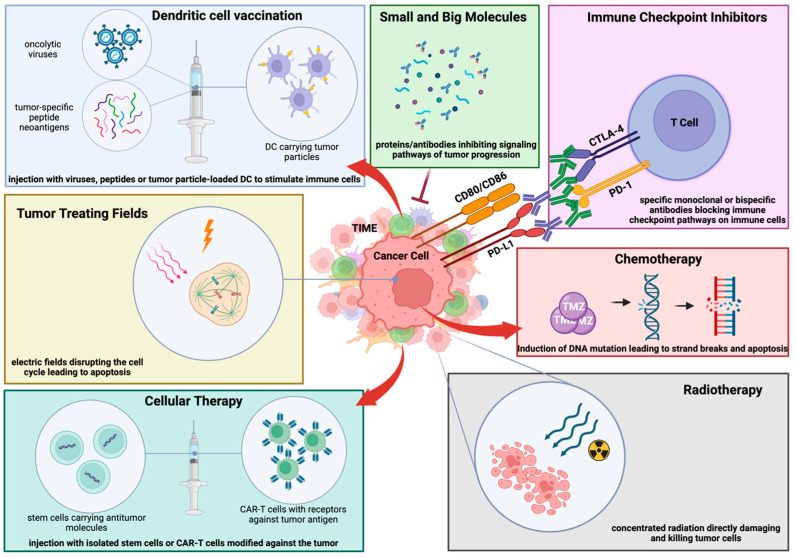
Overview of therapeutic approaches in GBM. Despite multimodal therapies in GBM, survival benefits are still limited. Several therapeutic options have been developed in the past decade, including immunotherapeutic interventions. Blue: Dendritic Cell Vaccination; Yellow: Tumor Treating Fields; Dark Green: Cellular Therapy; Grey: Radiotherapy; Red: Chemotherapy; Purple: Immune Checkpoint Inhibitors; Light Green: Small and Big Molecules. Created with BioRender.com (accessed on 24 January 2025).

**Table 1 cancers-17-00817-t001:** Summary of GBM treatment options with exemplary agents used either in clinical trials or already in active treatment. Agents and trials here include the most known and common agents and are just exemplary.

Therapy	Molecules/Agents	Function	Trials
Surgical resection		Acute reduction of tumor mass and alleviation of (neurological) symptoms	
Radiotherapy		Destruction of cancer cells and shrinkage of tumor mass post-surgery	
Chemotherapy and combinatorial approaches	TMZ, TMZ + Radiotherapy, TMZ + targeted inhibitors	Interference of DNA replication via alkylating agents	NCT00943826 [199] NCT00689221 [200] NCT00813943 [201] NCT00884741 [202]
Immune Checkpoint Inhibition	Nivolumab, Pembrolizumab, Ipilimumab	Enhancement of immune response against GBM	NCT03233152 [203] CA209-9UP [204] CAN-2409 [205]
Dendritic cell vaccination	Autologous DC pulsed with tumor lysate, peptides targeting EGFRvIII or WT1		DCVax [206] CDX-110 [207] GlioVax [208] SurVaxM [209]
Anti-angiogenesis (monoclonal antibodies)	Bevacizumab	Inhibition of blood vessel formation for restriction of tumor growth and nutrition	CA209-9UP [204]
Targeted inhibition (small molecules)	PI3K (Idelalisib), MEK (Trametinib), EGFR (Erlotinib)	Inhibition of molecular pathways to prohibit tumor progression	N0177 [210] most models only in research (in vitro)
Adoptive Cell Therapy	CAR-T cells, stem cells	Efficient delivery of therapeutic agents directly to tumor region	NCT05627323 [211] NCT02208362 [212] NCT03657576 [213]
Nanoparticles	Delphinidin, artificial nanocarriers	Enhancement of drug delivery via BBB	So far only in research (in vitro)
Tumor-treating fields (TTF)		Usage of electric fields for disruption of tumor cell division	NCT00379470 [144] NCT02831959 [214]

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
