# Peer review of "Immune Cell Interplay in the Fight Against GBM"

_cancers, 2025, doi:10.3390/cancers17050817_

Round 1

Reviewer 1 Report

Comments and Suggestions for Authors

This is a well-written and comprehensive review about the immunosuppressive microenvironment in Glioblastoma multiforme and the available therapeutic options. 

There are some minor issues that should be addressed by the authors:

The punctuation should be checked. There are quite often spaces or dots missing, especially where references have been inserted or at the end of sentences. That disturbs the reading flow.

The abbreviation GBM is just explained in line 124, although the term is used earlier in the review. 

I don't understand the meaning of the half sentence in line 99 "leads to tumor characteristics similar to cold tumors". You talk about cold tumors in this abstract and note in line 85 that MSI is present in them.

In the second part "The Tumormicroenvironment of GBM", it is noted in line 192/193 that "the recruitment and expansion of anti-inflammatory immune cells leads to .... metastasis." I would be very careful about using that term so general here. Metastasis happens rarely in GBM.

In line 201 "M2 macrophages" are mentioned but the difference to M1 macrophages is explained much later. To include a short explanation earlier would help to clarify their significance.

In line 256/257 it is claimed that "TREG is (besides TAM) the most abundant population in the TIME of GBM." I accept that TREGs are elevated in GBM but that they are the second most abundant population in GBM is news to me. Can you add a reference for that?

In Figure 1A and 1B the grammar should be checked, additionally, not all abbreviations used are explained in the figure legend.

Author Response

Reviewer #1:
This is a well-written and comprehensive review about the immunosuppressive microenvironment in Glioblastoma multiforme and the available therapeutic options.

There are some minor issues that should be addressed by the authors:

1. The punctuation should be checked. There are quite often spaces or dots missing, especially
where references have been inserted or at the end of sentences. That disturbs the reading
flow.

 Thank you for bringing this disturbing fact to our attention. We have gone through the
manuscript carefully and added or changed the punctuations where it was wrong or
even missing.

2. The abbreviation GBM is just explained in line 124, although the term is used earlier in the
review.
 We have corrected our mistake here by introducing the full name “glioblastoma
multiforme” already in line 104 and changed the abbreviation in line (now) 127.

3. I don't understand the meaning of the half sentence in line 99 "leads to tumor characteristics
similar to cold tumors". You talk about cold tumors in this abstract and note in line 85 that
MSI is present in them.
 We thank the reviewer for pointing out this confusing sentence as we already describe
that MSI is a characteristic of cold tumors and therefore changed it for more clarity to
(now line 102-104): The presence of MSI, caused by mutations in mismatch repair
(MMR) genes or expression abnormalities often results in poor immune response.

4. In the second part "The Tumormicroenvironment of GBM", it is noted in line 192/193 that
"the recruitment and expansion of anti-inflammatory immune cells leads to .... metastasis."
I would be very careful about using that term so general here. Metastasis happens rarely in
GBM.
 Thank you for pointing this out. It is a mistake from our side as GBM is indeed a tumor
that shows low to no metastasis. The sentence should read as follows:
Line 202ff: Especially the recruitment and expansion of anti-inflammatory immune
cells leads to the formation of a highly immunosuppressive TIME and eventually to
tumor immune evasion and progression in GBM [43].

5. In line 201 "M2 macrophages" are mentioned but the difference to M1 macrophages is
explained much later. To include a short explanation earlier would help to clarify their
significance.
 Line 215ff: In contrast to pro-inflammatory M1 macrophages that help fighting of
the tumor, M2 macrophages secrete anti-inflammatory cytokines which contribute
to the influx of further immunosuppressive cells and thereby to tumor progression,
which will be described in more detail in the further context.

In line 256/257 it is claimed that "TREG is (besides TAM) the most abundant population
in the TIME of GBM." I accept that TREGs are elevated in GBM but that they are the
second most abundant population in GBM is news to me. Can you add a reference for
that?

 We thank the reviewer for this very relevant comment as our phrasing here is misguiding. Upon an additional suggestion of Reviewer #4 we have not only rephrased the sentence but also reorganized and condensated the whole paragraph.
Line 272ff: The presence of TREG in GBM is associated with a poor prognosis due to these immunosuppressive properties [71]. They are recruited by GBM-derived CCL22 and CCL2, which attract TREGs to the tumor region via CCR4 [73]. As an
abundant population in the TIME of GBM, they contribute significantly to its immunosuppressive character and are characterized by a CD3+/CD4+/CD25high/CD127-/low/FoxP3high/CD45RA-/CD45R0+/CCR7+/CD62L-/CTLA4+immunophenotype. The immunosuppressive nature of TREG is aconsequence of several mechanisms. For one, they inhibit the production of IL-2
and IFN-γ [74] and thereby switching the immune response from a tumor-directed cytotoxic TH1-mediated immunity to a TH2-mediated response [75,76]. In addition, IL-2 deprivation leads to the disruption of metabolic pathways in effector T cells
driving them towards apoptosis [74]. They are also high producers of IL-10, TGF-ß and IL-35, cytokines which inhibit effector T-cell proliferation and induce the (DC) maturation towards a tolerogenic phenotype, which in turn induces additional TREG
formation [71,75–77].

6. In Figure 1A and 1B the grammar should be checked, additionally, not all abbreviations used are explained in the figure legend.
We proof-read the figure legend and added the missing abbreviations. The updated legend is marked in yellow.
Figure 1. Immune cell interactions within the tumour microenvironment (TIME) of glioblastoma multiforme (GBM). Tumor-infiltrating leukocytes (TIL) can either act (A) anti-tumoral and co-stimulate each other to attack the tumor mass, or they can act as (B) immunosuppressive regulators that are actively manipulated by GBM cells, or the TIL can manipulate each other, to
evade, reduce or completely disable an active anti-tumoral immune response. DC: dendritic cells; TAA: tumor-associated antigens; TNF-α: tumor-necrosis factor; GM-CSF: granulocyte-macrophage colony-stimulating factor; BCKA: branched-chain ketoacids; ICAM-1: intracellular adhesion molecule-1; LPA: Lysophosphatidic acid; MDSC: myeloid derived suppressor cells; CTL:cytotoxic T cells; IFN: Interferon; MDSC: myeloid derived suppressor cells; TREG: regulatory T cells; TGF-β: tumor growth factor beta; EGF: epidermal growth factor; CSF-1: colony-stimulating factor-1; CCL2: chemokine ligand 2;
PD-L1: programmed cell death ligand-1. Created with BioRender.com (accessed on 24th January 2025).

Reviewer 2 Report

Comments and Suggestions for Authors

The manuscript of Datsi and Vallieri addresses the challenges of treating glioblastoma, highlighting the complex cellular mechanisms of GBM-imune cells interactions that promote GBM progression and resistance to conventional therapies. Additionally, it explores innovative approaches, such as new drugs, tumor-treating fields, and promising immunotherapies, including CAR-T cells and dendritic cell vaccines. The review clearly and objectively explains the challenges to be overcome in the treatment of glioblastoma. The review is a valuable contribution to the understanding of strategies to combat this aggressive tumor. Below I point out minor changes needed to improve the review in general.

1 - In line 38, the title of subtopic 1.1 “tumor categorization” does not seem to reflect exactly what is adressed in the text. It would be better to change to: “mechanisms of immunosurveillance in hot and cold tumors” or “characteristics of hot and cold tumors”.

2 - In lines 148 and 155-156, the authors discuss the IDH gene mutation in glioblastomas and their better prognosis. However, according to the latest WHO classification (Loui 2021), IDH-mutant gliomas are no longer classified as glioblastoma. Glioblastomas are only IDH WT. Therefore, I believe that briefly discussing the most current glioma classification can help to make the review more complete.

Author Response

The manuscript of Datsi and Vallieri addresses the challenges of treating glioblastoma, 
highlighting the complex cellular mechanisms of GBM-immune cells interactions that promote 
GBM progression and resistance to conventional therapies. Additionally, it explores innovative 
approaches, such as new drugs, tumor-treating fields, and promising immunotherapies, 
including CAR-T cells and dendritic cell vaccines. The review clearly and objectively explains 
the challenges to be overcome in the treatment of glioblastoma. The review is a valuable 
contribution to the understanding of strategies to combat this aggressive tumor. Below I point 
out minor changes needed to improve the review in general.

1. In line 38, the title of subtopic 1.1 “tumor categorization” does not seem to reflect exactly 
what is addressed in the text. It would be better to change to: “mechanisms of 
immunosurveillance in hot and cold tumors” or “characteristics of hot and cold tumors”.

We thank the reviewer for his valuable suggestion and therefore changed the title of the 
subtopic to “Characteristics of hot and cold tumors” (line 43) as suggested.

2. In lines 148 and 155-156, the authors discuss the IDH gene mutation in glioblastomas and 
their better prognosis. However, according to the latest WHO classification (Loui 2021), 
IDH-mutant gliomas are no longer classified as glioblastoma. Glioblastomas are only IDH 
WT. Therefore, I believe that briefly discussing the most current glioma classification can 
help to make the review more complete.

 Line 149ff: With the development of new technologies and the introduction of 
molecular biomarkers in the diagnostic process, tumors of the central nervous system 
(CNS), are being re-grouped in the new WHO classification improving their grading 
and the biomarkers even have a prognostic impact [31]. For GBM, key genetic changes 
include mutations or dysregulations in the epidermal growth factor receptor (EGFR), 
tumor protein P53 (TP53), phosphatase and tensin homolog (PTEN), telomerase reverse 
transcriptase (TERT), and O-6-methylguanine-DNA methyltransferase (MGMT). 
Whereas previously isocitrate dehydrogenase (IDH) mutant tumors were classified as 
grade IV gliomas, they are now classified as astrocytomas and GBM as IDH wild-type.
[28], [31].

 ïƒ  Line 165 … GBM and nowadays not classified as GBM any longer,….

Reviewer 3 Report

Comments and Suggestions for Authors

This review article by Vallieri N. and Datsi A. examines the mechanisms driving glioblastoma growth and explores emerging treatment strategies, including novel chemotherapies, angiogenesis inhibitors, immune checkpoint blockade, and CAR-T cell therapies.

This review provides a comprehensive and well-structured analysis of tumor categorization, with a strong emphasis on the immune system's role in cancer progression and treatment response. The inclusion of key molecular markers and immune mechanisms, such as immune checkpoint molecules and tumor-infiltrating leukocytes, enriches the discussion and enhances the scientific rigor of the text.

The section on glioblastoma multiforme (GBM) is particularly compelling, effectively capturing the tumor’s aggressive nature, genetic heterogeneity, and treatment challenges.

Overall, the clarity, depth, and scientific accuracy of this work make it a valuable contribution to the field of oncology.

A few points need to be addressed:

1.     There is some repetition of concepts throughout the text.  For instance, at page 6, line 256-294, the paragraph discusses immune suppression by TREGs multiple times and the various ways they inhibit the immune response. While it’s important to emphasize their role, a more concise explanation could help avoid redundancy.
Suggestion: Condense repeated sections and focus on summarizing key points in a more concise manner.

2.     Several sentences are lengthy and overly complex, making them difficult to read. For example, at page 2, line 43,"These neoplasms can obtain advantageous phenotypes, which maintain proliferative signals, replicative immortality and the evasion of growth suppressors" could be reworded for clarity.

Author Response

Reviewer #4:

This review article by Vallieri N. and Datsi A. examines the mechanisms driving glioblastoma 
growth and explores emerging treatment strategies, including novel chemotherapies, 
angiogenesis inhibitors, immune checkpoint blockade, and CAR-T cell therapies.

This review provides a comprehensive and well-structured analysis of tumor categorization, 
with a strong emphasis on the immune system's role in cancer progression and treatment 
response. The inclusion of key molecular markers and immune mechanisms, such as immune 
checkpoint molecules and tumor-infiltrating leukocytes, enriches the discussion and enhances 
the scientific rigor of the text.

The section on glioblastoma multiforme (GBM) is particularly compelling, effectively 
capturing the tumor’s aggressive nature, genetic heterogeneity, and treatment challenges. 

Overall, the clarity, depth, and scientific accuracy of this work make it a valuable contribution 
to the field of oncology.

A few points need to be addressed:

1. There is some repetition of concepts throughout the text. For instance, at page 6, line 256-
294, the paragraph discusses immune suppression by TREGs multiple times and the 
various ways they inhibit the immune response. While it’s important to emphasize their 
role, a more concise explanation could help avoid redundancy. Suggestion: Condense repeated sections and focus on summarizing key points in a more concise manner.

 We thank the reviewer for reading this paragraph carefully, identifying repetitive 
sequences, and bringing this to our attention. We have therefore reordered and 
condensed this paragraph as follows:
Line 270ff: The presence of TREG in GBM is associated with a poor prognosis due to these immunosuppressive properties [71]. They are recruited by GBM-derived CCL22 and CCL2, which attract TREGs to the tumor region via CCR4 [73]. As an abundant 
population in the TIME of GBM, they contribute significantly to its immunosuppressive character and are characterized by a CD3+/CD4+/CD25high/CD127-/low/FoxP3high/CD45RA-/CD45R0+/CCR7+/CD62L-/CTLA4+immunophenotype. The immunosuppressive nature of TREG is a consequence of several mechanisms. For one, they inhibit the production of IL-2 and IFN-γ [74]and thereby switching the immune response from a tumor-directed cytotoxic TH1-mediated immunity to a TH2-mediated response [75,76]. In addition, IL-2 deprivation 
leads to the disruption of metabolic pathways in effector T cells driving them towards apoptosis [74]. They are also high producers of IL-10, TGF-ß and IL-35, cytokines which inhibit effector T-cell proliferation and induce the (DC) maturation towards a tolerogenic phenotype, which in turn induces additional TREG formation [71,75–77].

2. Several sentences are lengthy and overly complex, making them difficult to read. For 
example, at page 2, line 43,"These neoplasms can obtain advantageous phenotypes, which 
maintain proliferative signals, replicative immortality and the evasion of growth 
suppressors" could be reworded for clarity.

 Thank you for pointing this out. We have tried to resolve some of the complicated 
and long sentences by adding additional punctuation. Especially the one pointed out 
here in line 48 was rephrased as follows: These neoplasms can obtain advantageous 
phenotypes, which maintain proliferative signals. Additionally they maintain a 
replicative immortality and the capacity to evade or block the activity of growth 
suppressors,
